# Heat transfer control using a thermal analogue of coherent perfect absorption

Ying Li [1,2,3,7✉], Minghong Qi[1,2,3,7], Jiaxin Li[4,5], Pei-Chao Cao[6], Dong Wang[1,2,3], Xue-Feng Zhu[6], Cheng-Wei Qiu [5✉] & Hongsheng Chen [1,2,3✉]

Recent investigations on non-Hermitian physics have unlocked new possibilities to manipulate wave scattering on lossy materials. Coherent perfect absorption is such an effect that enables all-light control by incorporating a suitable amount of loss. On the other hand, controlling heat transfer with heat may empower a distinct paradigm other than using thermal metamaterials. However, since heat neither propagates nor carries any momentum, almost all concepts in wave scattering are ill-defined for steady-state heat diffusion, making it formidable to understand or utilize any coherent effect. Here, we establish a scattering theory for heat diffusion by introducing an imitated momentum for thermal fields. The thermal analogue of coherent perfect absorption is thus predicted and demonstrated as the perfect absorption of exergy fluxes and undisturbed temperature fields. Unlike its photonic counterpart, thermal coherent perfect absorption can be realized for regular thermal materials, and be generalized for various objects.

[1] Interdisciplinary Center for Quantum Information, State Key Laboratory of Modern Optical Instrumentation, ZJU-Hangzhou Global Scientific and Technological Innovation Center, Zhejiang University, Hangzhou 310027, China. [2] International Joint Innovation Center, Key Lab. of Advanced Micro/Nano Electronic Devices & Smart Systems of Zhejiang, The Electromagnetics Academy of Zhejiang University, Zhejiang University, Haining 314400, China. [3] Jinhua Institute of Zhejiang University, Zhejiang University, Jinhua 321099, China. [4] School of Mechatronics Engineering, Harbin Institute of Technology, Harbin 150001, China. [5] Department of Electrical and Computer Engineering, National University of Singapore, Singapore 117583, Singapore. [6] School of Physics and Innovation Institute, Huazhong University of Science and Technology, Wuhan 430074, China. [7] These authors contributed equally: Ying Li, Minghong Qi. ✉email: eleying@zju.edu.cn; chengwei.qiu@nus.edu.sg; hansomchen@zju.edu.cn

Manipulating heat transfer has a fundamental importance in thermal energy utilization[1–3], thermal management[4,5], and infrared signal processing[6,7]. For this purpose, various kinds of thermal metamaterials have been designed[8,9], whose functionalities are often achieved by meticulously patterning their thermal conductivity ($\kappa$) distributions[10–13]. However, some fundamental limitations come with the approach. The availability of materials in nature strongly restricts the range of effective $\kappa$ that can be realized[14]. In addition, after being fabricated, it is very difficult to modify the functionality of the thermal metamaterial, because the material thermal conductivity can hardly be tuned without using phase transition[15–17] or mechanical motion[18–21]. As an exemplary illustration, consider the effect of thermal transparency[22,23]. For the heat transfer in a common background material with thermal conductivity $\kappa_0$, any object inside the background will generally distort the temperature field if its thermal conductivity $\kappa$ is different from $\kappa_0$. A landmark achievement of thermal metamaterial is to achieve the transparent effect by surrounding the object with a thermal cloak[10,11], which can be designed based on the transformation theory[10–12,24] or the direct solutions[14,25,26]. Either theoretical method requires an effective $\kappa(\mathbf{r},\kappa_0)$ that depends on both the position $\mathbf{r}$ and the background host material's $\kappa_0$. It will be intrinsically challenging, if not impossible, to adapt to a different functionality or a different background $\kappa_0$ in a post-fabrication fashion.

An alternative route is thus needed to break the restriction of effective parameters and increase flexibility. Recently, it was proposed that heat transfer systems could be a unique platform to study non-Hermitian physics[27–33], which is originally used to describe dissipative wave systems[34–36]. Under this perspective, the effective Hamiltonian is used as a new tool to design heat transfer systems and realize unconventional functionalities. Despite the progress, this method is only applicable to the time-evolution of isolated systems. To study the more common steady-state response of a system to external heat sources, another important tool—the scattering theory is needed.

A representative application of scattering theory on non-Hermitian wave systems is the coherent perfect absorption (CPA)[37–41] of electromagnetic (EM) waves on lossy materials. Basically, it is destructive interference between the scattered waves from multiple sources, which provides a convenient method to control light with light. It would be highly desirable to be able to control heat with heat, namely by introducing additional heat sources into the system to avoid the use of complex and fixed structures. However, since there is no thermal field propagation in steady-state heat diffusion, all momentum-related concepts like wavenumber, interference, and reflection are absent. The establishment of a thermal scattering theory is thus challenging and highly nontrivial.

Here, we propose a method to build a correspondence between heat transfer and EM wave scattering in different dimensions. It introduces an "imitated momentum" for steady-state heat diffusion which is transported along a pseudo time. The thermal analogue of CPA in photonics is thus discovered. It is further identified as the perfect absorption of the exergy flux[42,43]. Based on it, thermal transparency can be achieved for naturally occurring materials by simply choosing adequate thermal inputs. Our theory offers strong flexibility in heat transfer control and is expected to inspire much broader domains of research in diffusive processes.

## Results

**Thermal scattering theory.** For heat diffusion, we are mostly interested in the steady-state temperature fields $T(\mathbf{r})$ that follows

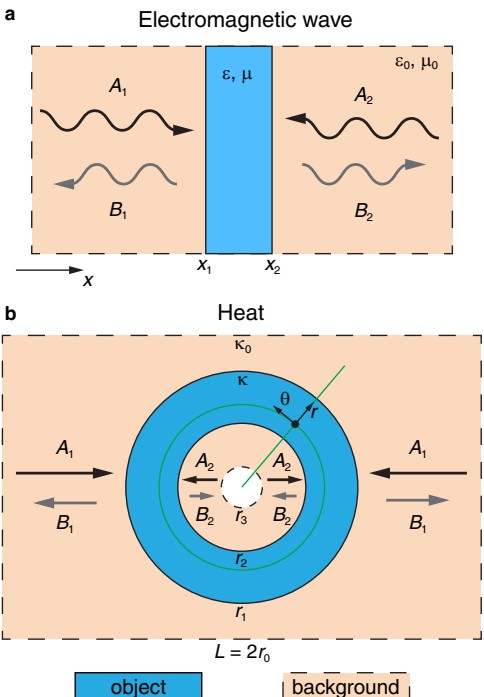

**Fig. 1 Correspondence between two distinct processes. a** Scattering of electromagnetic waves in one-dimension. **b** Steady-state heat diffusion in two-dimension. An object (light blue) is put in a background (beige). The black arrows represent incoming waves (fields) with amplitudes $A_1$ and $A_2$. The grey arrows represent outgoing waves (fields) with amplitudes $B_1$ and $B_2$. A polar coordinate system $(r,\theta)$ is built in **b** (green lines).

Fourier's law

$$\kappa \nabla^2 T = 0 \tag{1}$$

assuming a uniform $\kappa$. In one dimension, the equation is simply $T''(x) = 0$, and it is obvious that there is no directionality in the solution $T(x) = Ax$ (we set the constant term $T_0$ to zero throughout the theoretical derivation) because it cannot be decomposed into forward and backward parts.

To tackle the problem, we note that the essential difference between Eq. (1) and the governing equation for wave (Fig. 1a) comes from the time-harmonic oscillation of EM fields. Therefore, our idea is to add an auxiliary spatial dimension to Eq. (1) as pseudo time, and impose the temperature field to be periodic on it. As a concrete example (see Methods for more general discussions), consider the model in Fig. 1b, where the object is a circular ring with exterior and interior radius $r_1$ and $r_2$, respectively. The background is a rectangle with width $L = 2r_0$. The thermal conductivities of the object and background are $\kappa$ and $\kappa_0$. In the polar coordinate system $(r,\theta)$ with origin at the center of the object, the Fourier's law is now written as

$$\kappa \left( \frac{\partial^2 T}{\partial r^2} + \frac{1}{r}\frac{\partial T}{\partial r} + \frac{1}{r^2}\frac{\partial^2 T}{\partial \theta^2} \right) = 0 \tag{2}$$

It has a fundamental solution $T(r,\theta) = \mathrm{Re}[F(r)e^{im\theta}]$, here we focus on the common case with $m = 1$ (see Supplementary Note 1 for general results), which gives

$$F(r) = Ar + B/r, \quad r_2 \leq r \leq r_1 \tag{3}$$

$$F_1(r) = A_1 r/r_1 + B_1 r_1/r, \quad r \geq r_1 \tag{4}$$

$$F_2(r) = A_2 r_2/r + B_2 r/r_2, \quad r \leq r_2 \tag{5}$$

where the amplitudes $A$, $B$, $A_{1,2}$, and $B_{1,2}$ can be complex numbers to incorporate phases in the fields. A key operation here is to introduce a variable change: $r = e^{ikx} = e^x$ with a "wavenumber" $k = -i$, which is an imaginary number. Such an extended definition of wavenumber is commonly used in wave physics[44] to describe evanescent waves. By doing this, Eqs. (2) and (3)-(5) have the same form as for 1D wave scattering (see Supplementary Note 1 for the comparison). The two components of the field $F(x)$ appear to carry imitated momentums in the $x$ and $-x$ directions. The continuity of the temperature field and the heat flux require the following matching conditions

$$F_{1,2}(x_{1,2}) = F(x_{1,2}) \tag{6}$$

$$\kappa_0 F'_{1,2}(x_{1,2}) = \kappa F'(x_{1,2}) \tag{7}$$

where $x_{1,2} = \ln r_{1,2}$. Equations (6) and (7) have the same form as the matching conditions for the 1D EM wave scattering (see Supplementary Note 1 for the comparison). Therefore, it is natural for us to regard the original problem as 1D scattering in the $r$-direction, for which the $r$- and $r^{-1}$-components are the "forward" and "backward" parts, while $\theta$ is the pseudo time (Fig. 1b). Moreover, it can be shown that the two components actually carry exergy fluxes in the $-r$ and $r$ directions (see Supplementary Note 2). The exergy is a thermodynamic quantity defined as the maximum useful work a system can do by bringing it into thermodynamic equilibrium with the environment[41]. In our case, the useful work comes from the temperature difference between any local point in the system and the environment[42], meaning that one can extract work by putting a heat engine between them. Therefore, our decomposition of the temperature field gives important information about how the potentially useful thermal energy is distributed and transferred in the system.

We can study the thermal scattering problem through the transfer matrix $\mathbf{M}$, which is defined based on the amplitudes of the forward and backward fields:

$$\begin{pmatrix} B_2 \\ A_2 \end{pmatrix} = \mathbf{M} \begin{pmatrix} A_1 \\ B_1 \end{pmatrix} \tag{8}$$

The transfer matrix can be calculated by substituting Eqs. (3)-(5) into Eqs. (6) and (7)

$$\mathbf{M} = \begin{pmatrix} 1 & 1 \\ \kappa_0 & -\kappa_0 \end{pmatrix}^{-1} \begin{pmatrix} \cosh\Delta x & \frac{1}{\kappa}\sinh\Delta x \\ \kappa\sinh\Delta x & \cosh\Delta x \end{pmatrix} \begin{pmatrix} 1 & 1 \\ \kappa_0 & -\kappa_0 \end{pmatrix} \tag{9}$$

where $\Delta x = x_2 - x_1$. By comparing Eq. (9) with the transfer matrix for the electric fields in wave scattering, we see that the thermal conductivity $\kappa$ in heat diffusion corresponds to the admittance $Y = (\varepsilon/\mu)^{1/2}$ ($\varepsilon$ is the permittivity and $\mu$ is the permeability) for EM waves (see Supplementary Note 1). A rearrangement of Eq. (9) gives the scattering matrix $\mathbf{S}$ between the incoming and outgoing fields.

$$\begin{pmatrix} B_1 \\ B_2 \end{pmatrix} = \mathbf{S} \begin{pmatrix} A_1 \\ A_2 \end{pmatrix} = \frac{1}{\mathbf{M}_{22}} \begin{pmatrix} -\mathbf{M}_{21} & 1 \\ \det\mathbf{M} & \mathbf{M}_{12} \end{pmatrix} \begin{pmatrix} A_1 \\ A_2 \end{pmatrix}$$
$$= \begin{pmatrix} r_{11} & t_{12} \\ t_{21} & r_{22} \end{pmatrix} \begin{pmatrix} A_1 \\ A_2 \end{pmatrix} \tag{10}$$

The entries of $\mathbf{S}$ are the reflection ($r_{11}$ and $r_{22}$) and transmission ($t_{12}$ and $t_{21}$) coefficients, which are determined through the entries of $\mathbf{M}$. Since $\mathbf{M}_{12} = -\mathbf{M}_{21}$, the reflection coefficients of the scattering matrix $\mathbf{S}$ are equal ($r_{11} = r_{22}$). Despite of the imaginary wavenumber $k = -i$, we still have det $\mathbf{M} = \cosh^2\Delta x - \sinh^2\Delta x = 1$. Therefore, the heat transfer is reciprocal[45,46] ($t_{12} = t_{21}$). Together, the system preserves a parity symmetry, meaning that the temperature field is unchanged if we swap the exterior and interior parts of the object ($A_1 \leftrightarrow A_2$,

$x \leftrightarrow x_1 + x_2 - x$). This symmetry is unobvious because it is based on the logarithmic coordinate $x$. We explicitly write out $\mathbf{S}$ as

$$\mathbf{S} = \begin{pmatrix} \tilde{r} & \tilde{t} \\ \tilde{t} & \tilde{r} \end{pmatrix} = (\coth\Delta x - \cosh\gamma)^{-1} \begin{pmatrix} \sinh\gamma & \operatorname{csch}\Delta x \\ \operatorname{csch}\Delta x & \sinh\gamma \end{pmatrix} \tag{11}$$

where $\gamma = \ln(\kappa/\kappa_0)$.

**Thermal CPA and one-side CPA.** For the 1D wave scattering in Fig. 1a, when the material of the scatterer is lossless, it is easy to verify that the scattering matrix $\mathbf{S}_{\mathrm{EM}}$ satisfies $|\det \mathbf{S}_{\mathrm{EM}}| = 1$, which is a result of energy conservation. When the material is lossy, $n$ is a complex number, and $|\det \mathbf{S}_{\mathrm{EM}}| < 1$. Therefore, it is possible to add a suitable amount of loss into the material to meet the CPA condition $\det \mathbf{S}_{\mathrm{EM}} = 0$.

For heat transfer, the scattering matrix in Eq. (11) naturally satisfies $|\det \mathbf{S}| < 1$. Since the total exergy flux entering or leaving the object is proportional to the squared amplitude, $|\det \mathbf{S}| < 1$ means that the exergy is dissipated in the object, due to the local entropy generation (see Supplementary Note 2). Thermal CPA ($\det \mathbf{S} = 0$) can be reached by tuning $\gamma$ (which is a real number) to completely absorb the exergy flux entering the object. In addition, the magnitude of the heat flux will be $\theta$-independent under the condition (see Supplementary Note 2).

Setting $\det \mathbf{S} = 0$ gives

$$\kappa = \kappa_\pm{}^* = \kappa_0 \frac{r_1 \pm r_2}{r_1 \mp r_2} \tag{12}$$

$$\mathbf{S} = \frac{r_1 r_2}{r_1{}^2 + r_2{}^2} \begin{pmatrix} \mp 1 & 1 \\ 1 & \mp 1 \end{pmatrix} \tag{13}$$

The eigenvalues of $\mathbf{S}$ are $s_1 = 0$ and $s_2 = 2r_1 r_2/(r_1{}^2 + r_2{}^2)$. The thermal CPA condition is reached when the inputs meet the eigenvector $(A_1, \pm A_1)$ corresponding to $s_1$. Unlike in photonics that loss must be introduced, the thermal counterpart is readily realizable with normal materials. The inputs are simply symmetric or anti-symmetric, demonstrating the hidden parity symmetry of the system. The next question is how to generate the "incident" fields or input-1: $A_1 r/r_1$ and input-2: $A_2 r_2/r$. Heat transfer through a conductive system is usually studied by maintaining constant temperatures at the left and right sides ($T = \pm A_1 r_0/r_1$ at $r\cos\theta = \pm r_0$) and thermally insulating the upper and lower sides. If the object is absent, the temperature field is exactly the required $A_1(r/r_1)\cos\theta$, so the boundary condition can be regarded as the source of input-1 outside the object. On the other hand, consider a circle inside the object with radius $r_3 < r_2$. We apply a constant temperature distribution $T = A_2(r_2/r_3)\cos\theta$ on it. For this boundary condition, if the object is absent and the entire system is large (so that the effects of the outer boundaries are negligible), the temperature field outside $r_3$ will be $A_2(r_2/r)\cos\theta$, whose radial component meets input-2. Thus, the source for the input-2 inside the object is also found.

In photonics, the other eigenvector corresponding to the nonzero eigenvalue of $\mathbf{S}$ will lead to constructive interference of the scattered waves. Similarly, input fields that coincide with the other eigenvector corresponding to $s_2$ will lead to large outgoing fields. Note that this input can be realized by inverting the orientation of input-2. More generally, we can introduce a phase difference $\alpha$ between the two input fields in their $\theta$-dependences, such that $A_2 = \pm A_1 e^{i\alpha}$. The effects of such a phase difference are discussed in Supplementary Note 3 (also see Supplementary Fig. 1), where it is confirmed that the outgoing exergy flux is maximized at $\alpha = \pm\pi$. We also note that in photonics, CPA is the time-reversed process of lasing. One might thus be interested in the possibility of a thermal analogue of lasing[36]. However, the

time-reversed process of heat diffusion requires a material with negative thermal diffusivity as the gain medium, which has not been realized.

For CPA, the input fields from both sides are completely absorbed. In many cases such as thermal transparency and thermal cloaking, we are interested in the field outside the object. Therefore, we study the condition for no outgoing field on just one side of the object. We refer to this effect as one-side CPA, which is an analogue of unidirectional absorption in photonics. Since $B_2$ does not have to be zero, one may expect that the condition for one-side CPA is simply $B_1 = 0$. However, the field with nonzero $B_2$ will be "reflected" at the boundary of heat input-2, then "transmitted" through the object to contribute to an outgoing field in region $r \geq r_1$. Thus, we must consider all the outgoing fields after multiple scatterings and require that they are added to be zero.

To fully tackle the problem, we need to obtain the reflection coefficients at the two boundaries of heat inputs. We denote another scattering matrix $\mathbf{D}$ as

$$\begin{pmatrix} C_1 \\ C_2 \end{pmatrix} = \mathbf{D} \begin{pmatrix} B_1 \\ B_2 \end{pmatrix} = \begin{pmatrix} \widetilde{r}_a & 0 \\ 0 & \widetilde{r}_b \end{pmatrix} \begin{pmatrix} B_1 \\ B_2 \end{pmatrix} \quad (14)$$

where $C_1$ and $C_2$ are the amplitudes of the "reflected" fields outside and inside the object. The constant boundary conditions require that the boundary temperatures are not changed by this process, so $B_1 r_1/r_0 + C_1 r_0/r_1 = 0$ and $B_2 r_3/r_2 + C_2 r_2/r_3 = 0$. For simplicity, we have changed the actual boundary conditions at the left and right sides to the more symmetric condition ($T = A_1 r_0 \cos\theta/r_1$ at $r = r_0$). The reflection coefficients are $\widetilde{r}_a = -r_1^2/r_0^2$ and $\widetilde{r}_b = -r_3^2/r_2^2$. The total magnitudes $Z_1$ and $Z_2$ of the outgoing fields can be expressed as the infinite sequence

$$\begin{pmatrix} Z_1 \\ Z_2 \end{pmatrix} = \mathbf{S}' \begin{pmatrix} A_1 \\ A_2 \end{pmatrix} = (\mathbf{S} + \mathbf{SDS} + \mathbf{SDSDS} + \cdots) \begin{pmatrix} A_1 \\ A_2 \end{pmatrix} \quad (15)$$

The new scattering matrix $\mathbf{S}'$ can be explicitly solved by diagonalizing $\mathbf{SD}$

$$\mathbf{S}' = \begin{pmatrix} \widetilde{r} - \widetilde{r}_b(\widetilde{r}^2 - \widetilde{t}^2) & \widetilde{t} \\ \widetilde{t} & \widetilde{r} - \widetilde{r}_a(\widetilde{r}^2 - \widetilde{t}^2) \end{pmatrix} \quad (16)$$

The result is remarkably compact. We note that $\mathbf{S}'$ is still symmetric, but the parity symmetry is only preserved when $\widetilde{r}_a = \widetilde{r}_b$ (i.e., $r_1/r_0 = r_3/r_2$). The requirement is a geometric mirror symmetry for variable $x = \ln r$: $x_0 - x_1 = x_2 - x_3$. For CPA outside the object, $A_2 = A_1 \left[ \widetilde{r}_b \left( \widetilde{r}^2 - \widetilde{t}^2 \right) - \widetilde{r} \right] / \widetilde{t}$, which is even independent of the size of the background. Another great advantage of the one-side CPA condition is that it does not impose any restriction on the materials of the object and the background. A solution can be found for any values of $\kappa$ and $\kappa_0$. The limitations of conventional thermal metamaterials can thus be avoided, especially the fixed and restricted parameters.

**Numerical demonstration of thermal CPA and one-side CPA.** Based on the analytical results, we build a 2D finite-element model to verify and demonstrate the predicted effects. We first study the thermal CPA conditions in Eq. (12). The results are shown in Fig. 2. A Cartesian coordinate system ($w,h$) is built with its origin at $r = 0$ to facilitate the display. Under thermal CPA, there should be no outgoing field on both sides of the object, such that the temperature distributions outside and inside the object should meet input-1 and input-2, respectively. The temperature distributions in Fig. 2a, b meet this condition for both $\kappa > \kappa_0$ and $\kappa < \kappa_0$. For comparison, the temperature distributions with input-2 alone are simulated and plotted in Fig. 2c, d. The distribution for input-1 is not plotted, since it is simply a linear profile and

obviously meets the temperature profiles outside the object in Fig. 2a, b. For input-2, the reflection at the outer boundary is effectively removed by enlarging the size of the background to $L = 1$ m, thereby making the reflected field negligible in the displayed part. It is easy to identify the same pattern inside the object in Fig. 2a, b as the corresponding part of the input-2 fields in Fig. 2c, d (the other parts are made translucent). We further verify the effects by extracting the temperature distributions on the cut line $h = 0$ across the system and plot the results in Fig. 2e, f (scatters). In the intervals of the background (shaded by light orange), the numerical results overlap with the theoretical $r$-dependence for input-1 and input-2 (solid lines), so there is indeed no outgoing field with both incident fields perfectly absorbed.

The one-side CPA condition can be similarly studied with numerical simulations on the same model. Since it can be realized for any background and object material, we fix the thermal conductivity of the object $\kappa$ and choose two representative values of $\kappa_0 = \kappa/2.4$ and $\kappa/0.3$. According to Eq. (11), the reflection coefficients are $\widetilde{r} = -0.27$ and $0.37$; the transmission coefficients are $\widetilde{t} = 0.56$ and $0.5$. The ratio $\kappa/\kappa_0 = 2.4$ and $0.3$ do not match the CPA condition ($\kappa/\kappa_0 = 4.33$ and $0.23$). Therefore, the scattering cannot be completely suppressed on both sides. Thermal transparency outside the object is achieved when $A_2 = A_1 \left[ \widetilde{r}_b \left( \widetilde{r}^2 - \widetilde{t}^2 \right) - \widetilde{r} \right] / \widetilde{t} = 2.76$ K and $-3.52$ K, based on Eq. (15). The temperature distributions in these cases are plotted in Fig. 3a, b. The one-side CPA is confirmed that the fields outside the object are identical to the input-1 field in Fig. 2c, but the fields inside the object are different from the input-2 field in Fig. 2d. This is clearer by looking at the extracted temperature distributions on cutline $h = 0$ (Fig. 3c, d).

It is worth mentioning that a highly conductive background often requires an unreachable effective $\kappa$ for conventional thermal metamaterials to function[14], but our approach works for any values of $\kappa_0$ and $\kappa$. To further illustrate this, we plot in Fig. 3e the dependence of scattering coefficients $\widetilde{r}$ and $\widetilde{t}$, and the ratio $A_2/A_1$ required for one-side CPA on the ratio $\kappa/\kappa_0$. The abscissa is chosen to be $\gamma = \ln(\kappa/\kappa_0)$ to cover a wide range, and to demonstrate the antisymmetric (symmetric) dependence of $\widetilde{r}$ ($\widetilde{t}$) on it. The required input ratio is not symmetric due to multiple scatterings on the inner boundary. In particular, $A_2 = 0.1A_1 \neq 0$ for $\kappa = \kappa_0$ when the central void becomes the scatterer.

We note that the other one-side CPA condition for the field inside the object is also achievable, but one should be careful about the effects of the outer boundaries without rotational symmetry.

**Experimental demonstration of thermal CPA.** Our theoretical predictions can be experimentally realized with setups as shown in Fig. 4a, b (see Supplementary Fig. 4 for the photograph of the actual setup), where four aluminum heat sinks are used to maintain constant temperatures: $T_1$, $T_2$, $T_3$, and $T_4$. We use a copper bridge to connect the two central heat sinks and generate a linear temperature gradient on its top surface. To generate the desired input-2, the inner boundary of the background is made in contact with the copper bridge through a ring-shaped step with radius $r_3$. The orientation of the copper bridge can be rotated to introduce a phase into input-2 and thereby a phase difference between the two input fields.

The measured temperature distributions for the two CPA conditions are shown in Fig. 4c, d. It is easy to check that thermal transparency outside the object has been achieved in both cases. To confirm the thermal transparency inside the object, we plot the temperature distributions along a cutline at $h = 0$ (black dashed lines in Fig. 4c, d) in Fig. 4e, f. The measured temperatures

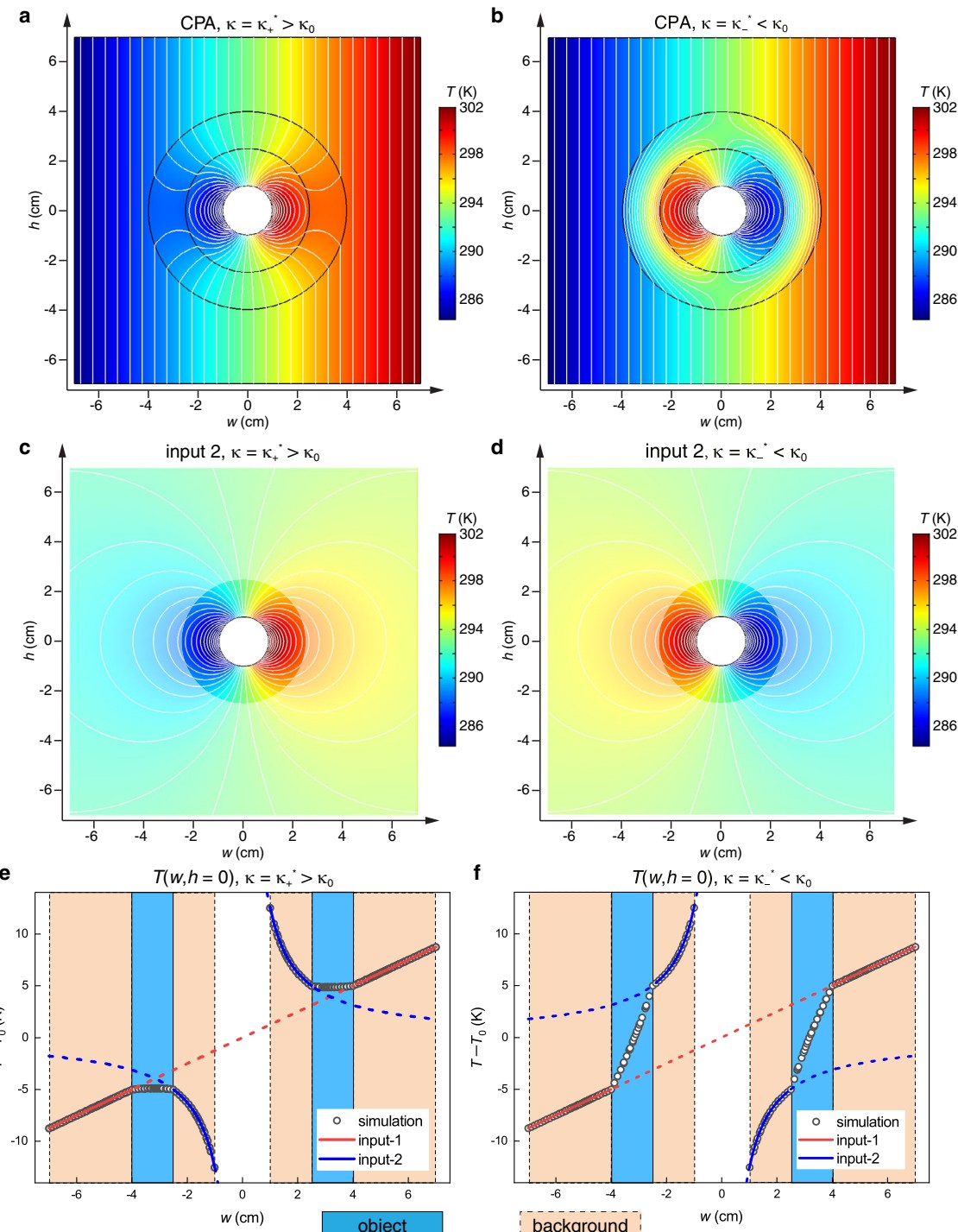

**Fig. 2 Thermal coherent perfect absorption (CPA).** The thermal conductivity $\kappa$ of the ring-shaped object is (**a**, **c**, **e**) larger or (**b**, **d**, **f**) smaller than that of the background $\kappa_0$. **a**, **b** The temperature distributions on the entire system with isothermal lines (white). **c**, **d** The temperature distribution of input-2 on a pure background that is much larger ($L = 1$ m) than the displayed part ($L = 14$ cm). **e**, **f** Temperature distributions along the line $h = 0$. The regions of the background (beige) and the object (light blue) are shaded. The incident fields input-1 (red) and input-2 (blue) are also plotted.

outside (inside) the object (scatters) are consistent with the input-1 (input-2) field (solid lines), indicating no outgoing field on either side of the object. The results are similar to the numerical results in Fig. 2e, f.

## Discussion

In conclusion, we propose a method to impart heat transfer processes with the concept of an "imitated momentum" and establish the theory of thermal scattering. The temperature field outside a ring-shaped normal material is found to be decomposed into two parts that carry exergy fluxes in opposite directions. The thermal scattering matrix for it is reciprocal and parity symmetric with real reflection and transmission coefficients, but the absolute value of its determinant is smaller than one due to entropy generation. It indicates the possibility to realize the thermal analogue of coherent perfect absorption (CPA). We numerically and experimentally verified the thermal CPA effect and

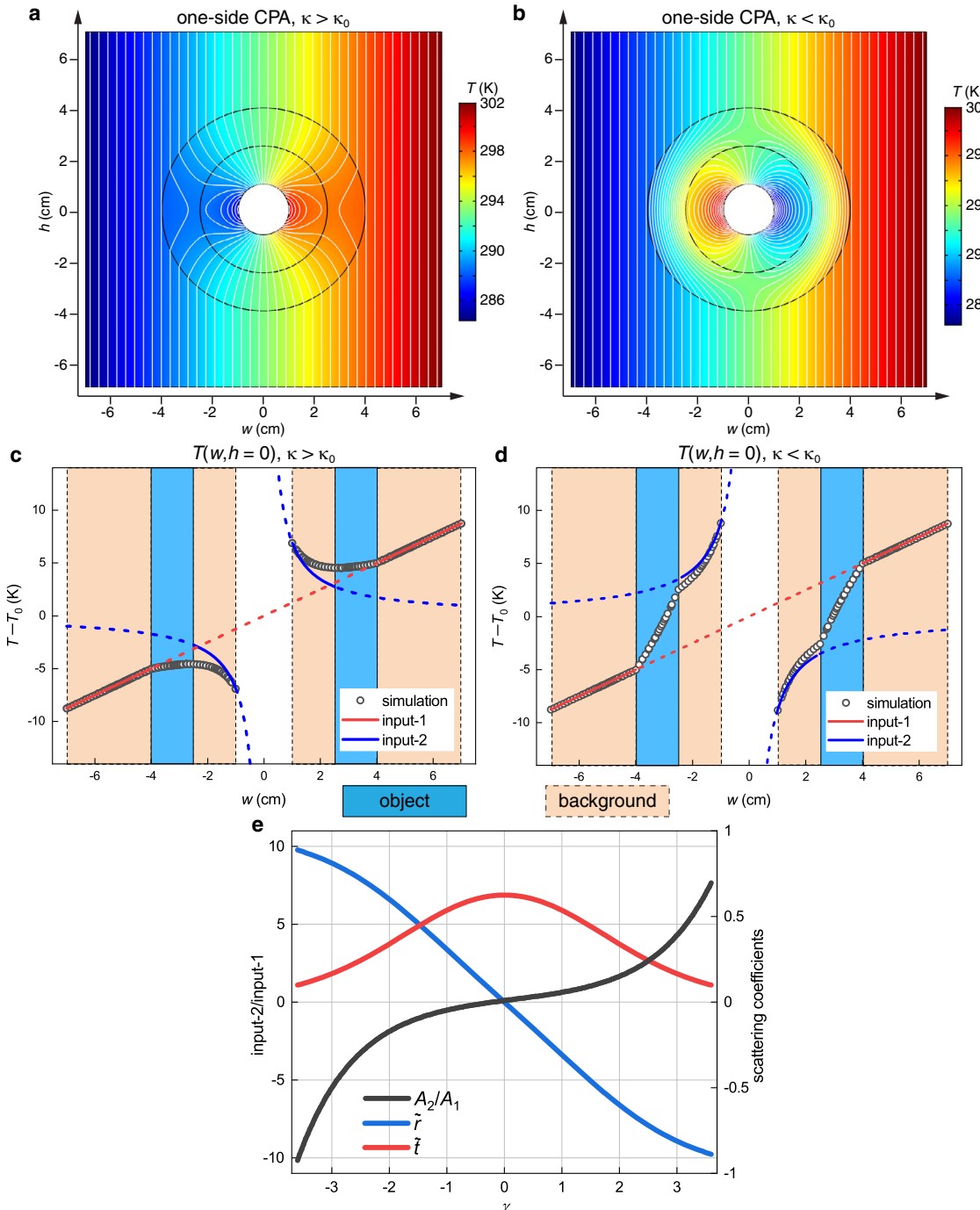

**Fig. 3 Thermal one-side coherent perfect absorption (CPA).** The ring-shaped object has an arbitrary thermal conductivity $\kappa$ that is (**a**, **c**) larger or (**b**, **d**) smaller than that of the background $\kappa_O$. **a**, **b** The temperature distributions on the entire system with isothermal lines (white). **c**, **d** Temperature distributions along the line $h = 0$. The regions of the background (beige) and the object (light blue) are shaded. The incident fields input-1 (red) and input-2 (blue) are also plotted. For one-side CPA only the field outside the object meets the input field. **e** The required inputs (black) and the reflection (blue) and transmission (red) coefficients for different combinations of the object and background materials.

demonstrated that it can be realized in various situations. The approach is suitable for thermal management applications with high flexibility and tunability. The work provides a powerful theoretical framework for studies on various diffusive processes not limited to heat transfer.

## Methods

**Constructing thermal scattering in general curvilinear systems.** For simplicity, we start from a 1D temperature field $T(\xi)$. Our auxiliary dimension can be chosen

as another spatial direction $\eta$ that is orthogonal to $\xi$. Considering the periodicity of $T(\xi,\eta)$ on $\eta$, the $\eta$ axis is generally a closed curve in physical space. We thus assume that $(\xi,\eta)$ form a 2D orthogonal curvilinear coordinate system, on which Eq. (1) can be explicitly written as

$$g^{11}\frac{\partial^2 T}{\partial \xi^2} + \left(-\frac{g^{11}}{2g_{11}}\frac{\partial g_{11}}{\partial \xi} + \frac{g^{22}}{2g_{22}}\frac{\partial g_{22}}{\partial \xi}\right)\frac{\partial T}{\partial \xi} + \left(\frac{g^{11}}{2g_{11}}\frac{\partial g_{11}}{\partial \eta} - \frac{g^{22}}{2g_{22}}\frac{\partial g_{22}}{\partial \eta}\right)\frac{\partial T}{\partial \eta} + g^{22}\frac{\partial^2 T}{\partial \eta^2} = 0$$

$$(17)$$

where we have eliminated $\kappa$. $g^{ij}$ and $g_{ij}$ are the contravariant and covariant components of the metric tensor, respectively. Now that we treat $\eta$ as pseudo time, the "time-harmonic" solution to Eq. (17) should have form $T(\xi,\eta) = \mathrm{Re}[F(\xi)e^{i\omega\eta}]$. If

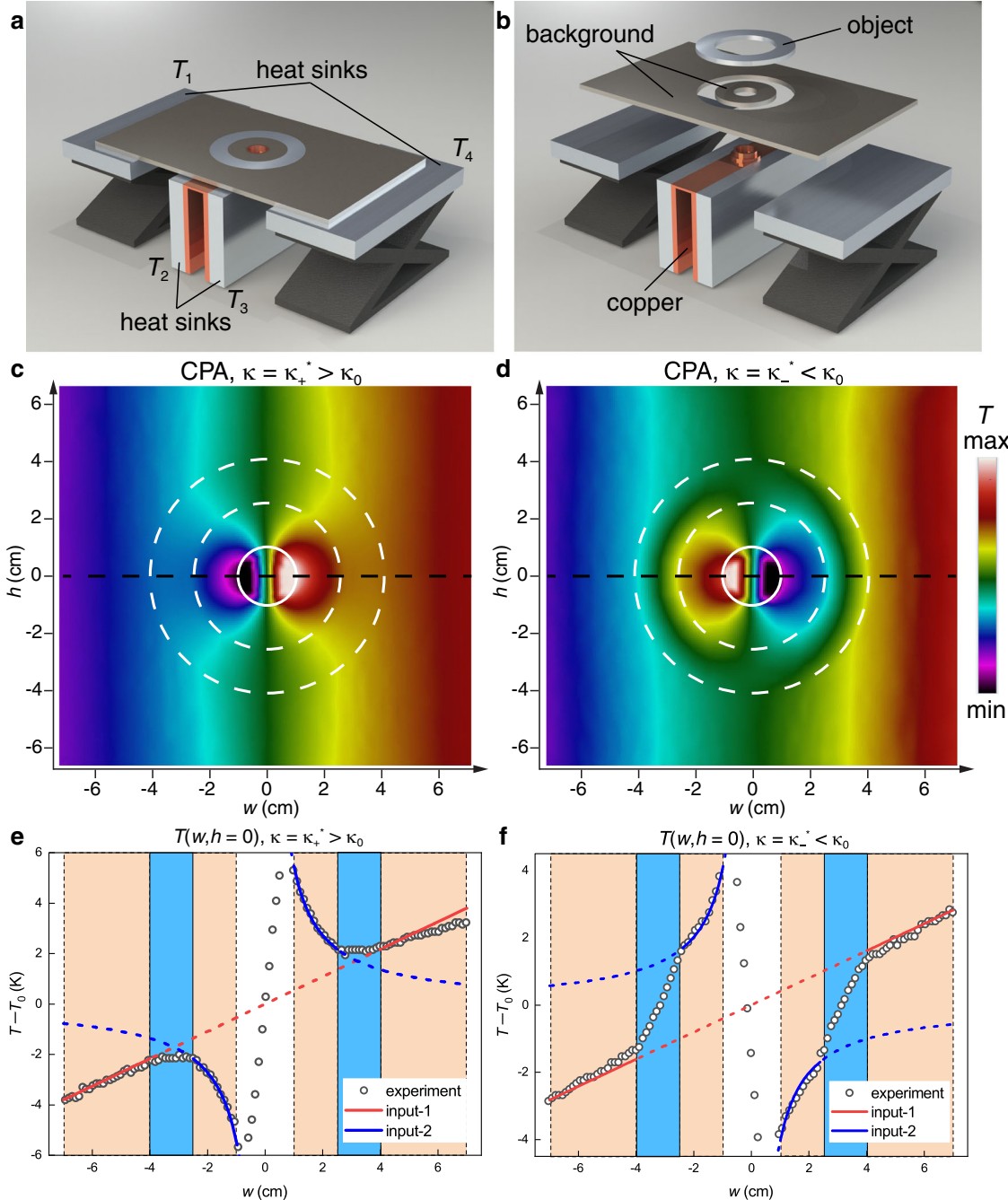

**Fig. 4 Experimental demonstration of thermal coherent perfect absorption (CPA). a, b** Schematics of the assembled (**a**) and disassembled (**b**) experimental setup. **c, d** Measured temperature profiles of the system when the thermal conductivity of the object (indicated with white dashed lines) $\kappa$ is (**c**) larger and (**d**) smaller than that of the background $\kappa_0$. **e, f** Measured temperature distributions along the line $h = 0$ (black dashed lines in **c** and **d**). The regions of the background (beige) and the object (light blue) are shaded. The incident fields input-1 (red) and input-2 (blue) are also plotted.

such a solution exists, $F(\xi)$ satisfies

$$g^{11}F'' + \left(-\frac{g^{11}}{2g_{11}}\frac{\partial g_{11}}{\partial\xi} + \frac{g^{22}}{2g_{11}}\frac{\partial g_{22}}{\partial\xi}\right)F' + \left[\left(\frac{g^{11}}{2g_{22}}\frac{\partial g_{11}}{\partial\eta} - \frac{g^{22}}{2g_{22}}\frac{\partial g_{22}}{\partial\eta}\right)i\omega - g^{22}\omega^2\right]F = 0 \tag{18}$$

where the coefficients must be independent of $\eta$. As a second-order ordinary differential equation, Eq. (18) has two linearly independent solutions $f(\xi)$ and $g(\xi)$. The trivial constant solution can be excluded if the coefficient before $F$ is nonzero. It is thus possible to construct the "forward" and "backward" fields in $\xi$ direction through different linear combinations of $f(\xi)$ and $g(\xi)$.

We will then treat the problem of 2D heat transfer through an object as a 1D scattering problem. It is assumed that the object and the background are isotropic and homogeneous with thermal conductivities $\kappa$ and $\kappa_0$. Also, the shape of the object is defined by $[\xi_1, \xi_2]$. Combining the general solution $F(\xi)$ with the matching

conditions, we can formally calculate the transfer matrix **M** and scattering matrix **S**, just like for Eqs. (3)-(7). The results are physically meaningful only when the forward (backward) field corresponds to the real input. We thus require them to meet with the steady-state field (up to a scaling factor) on the background when heat is launched from one side $\xi < \xi_1$ ($\xi > \xi_2$) and both the object and the boundary of the background on the other side are absent. According to the detailed ways of launching heat, one choice of the forward and backward fields on one side may be unsuitable on the other side, where different linear combinations of $f(\xi)$ and $g(\xi)$ must be used (for example, see Supplementary Note 4 for the case of an elliptic object).

**Numerical simulations.** The parameters are set as $r_0 = L/2 = 7$ cm, $r_1 = 4$ cm, $r_2 = 2.5$ cm, and $r_3 = 1$ cm. The height of the background is also $L$. The temperature boundary conditions are applied around temperature $T_0 = 293.15$ K. The

magnitude of input-1 is $A_1 = 5$ K. Therefore, the right and left sides of the background are maintained at constant temperatures $T_0 \pm A_1 r_0/r_1 = 301.9$ K and 284.4 K. The upper and lower boundaries are thermally insulated. Steady-state simulations were performed with COMSOL Multiphysics. For thermal CPA, the background thermal conductivity is set as $\kappa_0 = 90$ W m$^{-1}$ K$^{-1}$, which gives $\kappa = \kappa_+^* = 390$ W m$^{-1}$ K$^{-1}$ with $A_2 = A_1 = 5$ K and $\kappa = \kappa_-^* = 20.769$ W m$^{-1}$ K$^{-1}$ with $A_2 = -A_1 = -5$ K. For thermal one-side CPA, the thermal conductivity of the object is $\kappa = 120$ W m$^{-1}$ K$^{-1}$. The background thermal conductivity is set as $\kappa_0 = 50$ W m$^{-1}$ K$^{-1}$ and 400 W m$^{-1}$ K$^{-1}$.

**Experiments.** The system has the same geometry as in the numerical simulations with a thickness of 2 mm. For the first CPA condition, the background is carbon steel with $\kappa_0 = 50$ W m$^{-1}$ K$^{-1}$. The object is aluminum with $\kappa = 217$ W m$^{-1}$ K$^{-1}$. The temperatures of the left and right edges of the background are maintained at $T_1 = 296.65$ K and $T_4 = 303.65$ K, respectively. The temperatures of the left and right copper pillars are maintained at $T_2 = 284.15$ K and $T_3 = 316.15$ K, respectively. The central temperature is $T_0 = 300.15$ K. For the second CPA condition, the object is carbon steel, and the background is aluminum. The temperatures are $T_1 = 298.15$ K, $T_2 = 304.15$ K, $T_3 = 283.15$ K, and $T_4 = 319.15$ K. The central temperature is $T_0 = 301.15$ K. The entire system is covered by a thin polypropylene film to ensure a high thermal emissivity (0.97) for the measurement with an infrared camera.

## Data availability

Data presented in this publication is available on Figshare with the following identifier.
https://doi.org/10.6084/m9.figshare.19242786.v1

## Code availability

The codes used in the current study are available from the corresponding authors upon reasonable request.

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

## Acknowledgements

The work was sponsored by the National Natural Science Foundation of China (NNSFC) under Grants No. 92163123 (Y.L.), No. 61625502 (H.C.), No. 11961141010 (H.C.), and No. 61975176 (H.C.), the Top-Notch Young Talents Program of China (H.C.), the Fundamental Research Funds for the Central Universities (H.C.), and Ministry of Education, Republic of Singapore, via grant No.: R-263-000-E19-114 (C.-W.Q.).

## Author contributions

Y.L. conceived the idea. Y.L. constructed the theory and performed the numerical simulations. Y.L., M.Q., J.L., and C.P.C. designed the experiments. Y.L., M.Q., and D.W.

performed the experiments. Y.L. wrote the manuscript. All the authors contributed to the manuscript editing. Y.L., C.W.Q., and H.C. supervised the work.

## Competing interests

The authors declare no competing interests.
