## [Peer Review File · Nature Communications]

REVIEWER COMMENTS

Reviewer #1 (Remarks to the Author):

In this paper authors propose a way to conceptually control and tune the heat transfer processes by introducing imitated momentums and thus relaying on the theory of wave scattering in the long wavelength limit that they recall as thermal scattering.

Analogy is done in term of incident and scattered part, and they show some examples of application. The general approach is correct from the technical point of view. The paper is in an acceptable format for publication.

Few minor remarks

- In the quasi-static case the wavelength is quasi infinite and thus it is not easy to define the source position in an experiment in the sense of an incident field but rather a total field. How would you manage this?
- It would be nice to have the image of the experimental part in the text.

Reviewer #2 (Remarks to the Author):

This is a theoretical and experimental paper looking into how coherent perfection absorption (CPA) -- a phenomenon in which light waves can perfectly cancel each other through a combination of loss and interference -- can be transferred to heat flow. This is not a straightforward generalization, since heat does not flow in waves, so heat signals do not interfere in the same way as light (or other waves). The authors' idea is to look at the specific case of a 2D cylindrically symmetric shell, and redefine "forward"/"backward" heat signals as heat gradients that scale as r and $1/r$ (where r is the radial coordinate in polar coordinates). With this redefinition, one can define a "transfer matrix" and "scattering matrix" and examine the conditions under which "perfect absorption" occurs.

The heat CPA condition is that (i) the heat signal from infinity has a pure r scaling (with no $1/r$ part), similar to if the shell and origin heat source are absent, and (ii) the heat signal from the origin has a pure $1/r$ scaling (with no r part), as though the shell and heat source at infinity are absent. The authors also study a "one-sided CPA" case where only one of these conditions holds.

The concept is mathematically sound, and the authors back it up with experiments that seem to be technically sound. As far as publication in Nature Communications, however, what gives me pause is whether this notion of "perfect absorption" is relevant and general enough.

There are a few related doubts:

1. Does the "no reflection" condition that the authors achieve have physical meaning? In the original CPA context, the incident and outgoing energy fluxes have physical meaning, and achieving CPA means that the incident energy flux enters the body and is completely dissipated. In the current context, the

directions are interpreted in terms of r vs $1/r$ scaling, but what is the relationship with the physical heat flux?

1a. Put another way, for the optical case, the incident and outgoing waves can in principle be separated using beamsplitters, so there's an experimental way of discerning the incident and outgoing flux. If one wants to tell whether there is any outgoing "heat signal", how would that be done? Is checking the r -scaling (curve fitting) the only way?

1b. Another related question: what does $|\det(S) < 0|$ mean physically? The authors seem to say some words about dissipation but it's never very concrete, maybe because the connection to heat flux or other physical quantities hasn't been developed.

2. The CPA concept is not just about cloaking; one of the key behaviors of interest is that adjusting the incident illumination, one can switch between weak and perfect transfer of energy into a body. Notably, the relative phase of the incident beams can switch between the two cases. Does any aspect of this survive in the heat case? It's hard to see how, since there is no concept of a phase in the heat case. Without the ability to have adjustable incident signals, how can one justify calling this "controlling heat with heat" by analogy with CPA?

3. The approach seems to be very specific to cylindrically symmetric shells and polar coordinates. Can it be generalized to more arbitrary geometries, and if so how? If this is possible, some brief comments would be helpful. It's worth noting that the generalization of CPA to higher dimensions and more complicated geometries is easy, once the 1D case is understood.

3a. On a related note, the authors focus on the $\cos(\theta)$ harmonic, but is this necessary? What about the other harmonics?

I would have to see the answers to these questions before being able to determine whether the paper is suitable for publication in Nature Comms.

Reviewer #3 (Remarks to the Author):

A wave scattering based approach to understanding thermal diffusion, and for regulating thermal transport, has been proposed. A related aim involves the adapting and obtaining of novel phenomena in thermal conduction, such as coherent perfect absorption (CPA), thus far associated with coherent optics.

Generally, while the objective/s are interesting the methodology as well as the presentation are both weak

At the very outset, much intriguing terminology, not typically associated with heat transfer such as an "imitated momentum" and "pseudotime" has been indicated. The temperature variation has been then considered to be periodic in the "pseudotime" dimension. It is unclear what is the physical meaning

behind such a “pseudotime”. Other aspects that are unclear with respect to the usage in describing real heat transfer include the following: , e.g.,

(1)What is the reasoning behind substituting for the thermal conductivity an inverse permeability?

(2)Also, from Figure 1, is there supposed to be some correspondence between “ n, μ ” and “ k ”?

(3)In Eqn. (4) is there a flux matching being considered? How then would the flux be modeled in terms of wave scattering? Is there a corresponding interpretation? What role would the imaginary time play in the definition of the flux?

(4)What is the bandwidth of the supposed CPA? Would there be associated laser-like characteristics given that the CPA is considered somewhat as a time-reversed laser.

(5)An intriguing aspect then is how exactly time reversal symmetry (TRS) is being violated- as indicated on page 6. Firstly, some kind of a “pseudotime” is being used. Is the TRS related to this pseudotime or regular/ordinary time? It has also been rather glibly stated that “Together, the system preserves a parity symmetry which is not obvious from its geometry”? How could such a statement be more clearly justified?

(6)Also, the authors indicate occasionally that there is “no thermal correspondence” – on page 8, as phase considerations are irrelevant?

(7)In the numerical simulation, what is the reason for the choice of the “representative values”? How do the temperature profiles in Figs. 2e and 2f or Figs. 3c or 3d. accord with a coherent aspect?

There are also rather sweeping statements such as “Only the complete annihilation of two opposite fields is nontrivial” which are quite confusing in scope. What is also a “one side CPA”? Is there an optical analog and

From the experimental point of view, the results in, e.g., Figures 4c and 4d, are intriguing, in terms of a “dipole-like” characteristic. It is unclear how these results are in accord with the simulations indicated previously. The authors should clearly explain the similarities and differences and why CPA is obvious to them.

In summary, considering the lack of a clear definition of a phase, it is difficult to understand the related coherence aspect in the invoked CPA. The use of a pseudotime needs much more clear explanation. In summary, the authors should reconsider the physical premise/s behind their work and truly aim to indicate the utility of their approach for obtaining insights into thermal transport.

** See Nature Research's author and referees' website at www.nature.com/authors for information about policies, services and author benefits.

Reply to Reviewer #1:

In this paper authors propose a way to conceptually control and tune the heat transfer processes by introducing imitated momentums and thus relaying on the theory of wave scattering in the long wavelength limit that they recall as thermal scattering.

Analogy is done in term of incident and scattered part, and they show some examples of application.

The general approach is correct from the technical point of view. The paper is in an acceptable format for publication.

Our reply: We thank the Referee for the positive comments.

Few minor remarks

- In the quasi-static case the wavelength is quasi infinite and thus it is not easy to define the source position in an experiment in the sense of an incident field but rather a total field. How would you manage this?

Our reply: Thank you for pointing out the issue. Indeed, the wavelength will approach infinity in the quasi-static case. In our case, this corresponds to a vanishing wavenumber k which makes the incident field ill-defined. However, since the pseudo time is actually the angle θ , the pseudo time periodicity should be $2m\pi$. It follows that $k = -mi$, which cannot be made arbitrarily small to reach quasi-static case. The only exception is when the field is θ -independent such that $k = 0$. In that case, it is indeed impossible to distinguish an incident field from the total field. It exactly reflects the original difficulty in steady-state heat diffusion, but is a special and trivial case where the field amplitude is zero.

- It would be nice to have the image of the experimental part in the text.

Our reply: We thank the referee for the suggestion. A photo of the experimental setup was included in the Supplementary Information as Fig. S4.

Reply to Reviewer #2:

This is a theoretical and experimental paper looking into how coherent perfection absorption (CPA) -- a phenomenon in which light waves can perfectly cancel each other through a combination of loss and interference -- can be transferred to heat flow. This is not a straightforward generalization, since heat does not flow in waves, so heat signals do not interfere in the same way as light (or other waves). The authors' idea is to look at the specific case of a 2D cylindrically symmetric shell, and redefine "forward"/"backward" heat signals as heat gradients that scale as r and $1/r$ (where r is the radial coordinate in polar coordinates). With this redefinition, one can define a "transfer matrix" and "scattering matrix" and examine the conditions under which "perfect absorption" occurs.

The heat CPA condition is that (i) the heat signal from infinity has a pure r scaling (with no $1/r$ part), similar to if the shell and origin heat source are absent, and (ii) the heat signal from the origin has a pure $1/r$ scaling (with no r part), as though the shell and heat source at infinity are absent. The authors also study a "one-sided CPA" case where only one of these conditions holds.

The concept is mathematically sound, and the authors back it up with experiments that seem to be technically sound. As far as publication in Nature Communications, however, what gives me pause is whether this notion of "perfect absorption" is relevant and general enough.

Our reply: We are grateful to the Referee for the accurate summarization and positive comments on our work. We also appreciate the suggestion to give more discussion on the "perfect absorption" in heat diffusion, which we confirm as a physically relevant and general concept.

There are a few related doubts:

1. Does the "no reflection" condition that the authors achieve have physical meaning? In the original CPA context, the incident and outgoing energy fluxes have physical meaning, and achieving CPA means that the incident energy flux enters the body and is completely dissipated. In the current context, the directions are interpreted in terms of r vs $1/r$ scaling, but what is the relationship with the physical heat flux?

Our reply: Thank you for the question. The perfect absorption in heat diffusion has physical meanings. In the revised Supplementary Information (SI), we have added a note (Supplementary Note 2) to give two interpretations.

First, we define a thermoelectric potential $TEP = (T - T_0)^2/2$ which is proportional to the power that can be generated by a thermoelectric generator connecting the local point and the ambient environment (at temperature T_0). We then identify the TEP flux as $f = -D\nabla T$. The total TEP flux through a circle outside the object is then

$$Q_1 = -\pi D \left[A_1^2 e^{2(x-x_1)} - B_1^2 e^{-2(x-x_1)} \right] \quad (1)$$

Similarly, that in the interior of the object is

$$Q_2 = -\pi D \left[B_2^2 e^{2(x-x_2)} - A_2^2 e^{-2(x-x_2)} \right] \quad (2)$$

Therefore, the incoming and outgoing fields have physical implications as their amplitudes determine the TEP flux towards and away from the object. The perfect absorption condition means that, similar as in wave dynamics, the TEP flux that enters the object is completely dissipated. Unlike in wave physics, the TEP flux is also dissipated along the route, so Q varies with x or r . It reflects the dissipative nature of any thermal material.

Second, we may also identify the perfect absorption condition by using the heat flux. It can be shown that the heat flux magnitude $|\mathbf{q}| = \kappa r^{-1}(T_x^2 + T_\theta^2)^{1/2}$ will be θ -independent or isotropic under the condition.

In summary, the concept of perfect absorption is physically related with temperature differences, either between the system and the environment (the first) or inside the system (the second). In addition to the potential to generate thermoelectric power, the temperature difference may also be used for sensing (*Adv. Mater.* **27**, 7752) or infrared signal processing (*Mater. Today* **45**, 120). The concept is thus both physically relevant and of general interests. We have included the discussion in the revised manuscript (page 5 and page 7).

1a. Put another way, for the optical case, the incident and outgoing waves can in principle be separated using beam splitters, so there's an experimental way of discerning the incident and outgoing flux. If one wants to tell whether there is any outgoing "heat signal", how would that be done? Is checking the r -scaling (curve fitting) the only way?

Our reply: We thank the referee for bringing out the comparison between the optical and thermal cases. It is equally easy (or difficult) to experimentally check the existence of an optical or thermal outgoing field. Consider the following Fig. 1, where optical and thermal CPA conditions are achieved in Fig. 1a and c. In the optical case (Fig. 1b), one can put a beam splitter (red) in front of the object. Although there was originally no outgoing wave, the beam splitter itself blocks half of the incident wave 1, and breaks the CPA condition. As a result, one receives signal in the upper direction, but cannot determine whether there was outgoing wave. In the thermal case (Fig. 1d), one can put a heat flux meter outside the object. Similarly, the device itself alters the original temperature field which has no gradient in the vertical direction, so the measured flux in the vertical direction does not imply the existence of outgoing field.

The reliable method to discern the incoming and outgoing components might still be direct field probing. For heat transfer, it is not so hard to measure the entire field with an infrared camera as we did in this work. Then we can indeed perform a curve fitting to accurately separate the two fields. An easier way to determine whether there is no outgoing field is to simply measure the temperatures T_a and T_b at two points along a vertical line as in Fig. 1c. If they are outside the object, the condition is $T_a = T_b$. If they are inside the object, the condition is $(T_a - T_0)r_a^2 = (T_b - T_0)r_b^2$, where $r_{a,b}$ are the radial coordinates of the two points. We have included the discussion in the revised manuscript (page 10).

Figure 1. Schemes to detect the outgoing fields

1b. Another related question: what does $|\det(S)| < 1$ mean physically? The authors seem to say some words about dissipation but it's never very concrete, maybe because the connection to heat flux or other physical quantities hasn't been developed.

Our reply: We agree with the referee that the connection between $|\det S|$ and the heat flux has not been concretely discussed. In the revised version, we identify another physical flux—the TEP flux, as defined above. It is easy to see from Eqs. (1) and (2) that $|\det S| < 1$ means more TEP flux enters than leaves the object. This is always true for heat diffusion due to a dissipation term $D(\nabla T)^2$, as discussed in the revised SI. We have included the discussion in the revised manuscript (page 7).

2. The CPA concept is not just about cloaking; one of the key behaviors of interest is that adjusting the incident illumination, one can switch between weak and perfect transfer of energy into a body. Notably, the relative phase of the incident beams can switch between the two cases. Does any aspect of this survive in the heat case? It's hard to see how, since there is no concept of a phase in the heat case. Without the ability to have adjustable incident signals, how can one justify calling this "controlling heat with heat" by analogy with CPA?

Our reply: Thank you for reminding us the issue. The mentioned behaviors in optics all have their thermal counterparts. The amplitude and phase of the incident fields can be easily tuned through the temperatures and orientations of the heat sinks. In the conventional framework, there is indeed no concept of a phase in heat diffusion. This is exactly why we introduce the polar coordinate θ as a pseudo time. Thereby, we can naturally define the phase as the orientation of

the field, which can be written as the angle of the complex amplitude. We have included the discussion in the revised manuscript (page 8).

We have added a note (Supplementary Note 3) in the revised SI to discuss the effects of a relative phase α between the two input fields. Additional experiments were performed for $\alpha = -\pi/6$ (Fig. S1d). The measured temperature field confirms the sensitivity of the CPA effect to the relative phase. We predict and show that, starting from a CPA condition with $\alpha = 0$, more TEP flux flows out of the object as α increases. At $\alpha = \pm\pi$, the input fields meet the other eigenvector of the scattering matrix, with a maximized outgoing TEP flux (around 81% of the incoming flux). All the results meet well with the standard theory of CPA in optics. It supports our approach as a justified thermal analogy of CPA.

3. The approach seems to be very specific to cylindrically symmetric shells and polar coordinates. Can it be generalized to more arbitrary geometries, and if so how? If this is possible, some brief comments would be helpful. It's worth noting that the generalization of CPA to higher dimensions and more complicated geometries is easy, once the 1D case is understood.

Our reply: Thank you for the question. It is possible to generalize our approach to other geometries. In fact, in the Methods part of our manuscript, we have included a general discussion based on an arbitrary curvilinear system. Moreover, in the SI, we have included a detailed note on the CPA condition for an elliptic object (see Supplementary Note 4, Fig. S2, and Fig. S3). We believe the cases in higher dimensions and for more complicated geometries are equally probable as in optics, and look forward to future works on these topics.

3a. On a related note, the authors focus on the $\cos(\theta)$ harmonic, but is this necessary? What about the other harmonics?

Our reply: Thank you for the question. The reason why we focus on the $\cos\theta$ harmonic is simply that this is the most commonly encountered kind of temperature fields, where just one hot and one cold heat sinks exist. For higher harmonics like $\cos(m\theta)$, the same approach also applies. One only has to change the wavenumber k from $-i$ to $-mi$ (see Supplementary Note 1). The input fields must be generated by multiple alternative hot and cold heat sinks, which are rare in practice. We have included the discussion in the revised manuscript (page 5).

Reply to Reviewer #3:

A wave scattering based approach to understanding thermal diffusion, and for regulating thermal transport, has been proposed. A related aim involves the adapting and obtaining of novel phenomena in thermal conduction, such as coherent perfect absorption (CPA), thus far associated with coherent optics.

Generally, while the objective/s are interesting the methodology as well as the presentation are both weak.

Our reply: We thank the referee for commenting that our phenomena are novel and our objectives are interesting. To clarify our methodology and improve our presentation, we have largely revised our manuscript and added Supplementary Notes 1-3 to the Supplementary Information (SI).

At the very outset, much intriguing terminology, not typically associated with heat transfer such as an “imitated momentum” and “pseudotime” has been indicated. The temperature variation has been then considered to be periodic in the “pseudotime” dimension. It is unclear what is the physical meaning behind such a “pseudotime”.

Our reply: We are sorry that these phrases might bring confusions to the referee. They are used to indicate the mapping from the 2D heat diffusion to 1D wave scattering. In the revised Supplementary Information (SI), we have added a note (Supplementary Note 1) to detailly discuss this mapping. That the angular coordinate θ being called “pseudo time” is justified by the governing equation and the periodicity of the temperature field on it. After the correspondence has been established, we still interpret θ with its original physical meaning as the angular coordinate.

Other aspects that are unclear with respect to the usage in describing real heat transfer include the following: e.g.,

(1) What is the reasoning behind substituting for the thermal conductivity an inverse permeability?

Our reply: Thank you for the question. We apologize for the unclearness in the original manuscript. We did not mean to substitute the thermal conductivity with an inverse permeability, as they belong to different governing equations for different physical processes. What we meant is that the two equations have the same form by regarding the two material parameters as the same kind of coefficient. In the revised Supplementary Information (Supplementary Note 1), we have re-examined this correspondence and found it more appropriate to map the thermal conductivity to the admittance, based on the form of the transfer matrix. We have included the discussion in the revised manuscript (page 6).

(2) Also, from Figure 1, is there supposed to be some correspondence between “ n, μ ” and “ k ”?

Our reply: Thank you for the question. As mentioned above, the thermal conductivity κ is mapped to the admittance $Y = (\varepsilon/\mu)^{1/2}$ based on the form of the transfer matrix. Therefore, we have revised Fig. 1 to replace n with ε .

(3) In Eqn. (4) is there a flux matching being considered? How then would the flux be modeled in terms of wave scattering? Is there a corresponding interpretation? What role would the imaginary time play in the definition of the flux?

Our reply: Yes. In heat diffusion, the heat flux is $\mathbf{q} = -\kappa\nabla T$. Eq. (4) (the second line) is thus the matching of heat flux in the radial direction. In wave scattering, the corresponding matching condition (Eq. (S8) of the revised SI, second line) is for the continuity of the magnetic field \mathbf{H} , which is not a flux. The definitions of the heat flux and the magnetic field are both standard, where θ should be treated as the angular coordinate in calculating the temperature gradient. We clarified the matching condition in the revised manuscript (page 5).

(4) What is the bandwidth of the supposed CPA? Would there be associated laser-like characteristics given that the CPA is considered somewhat as a time-reversed laser.

Our reply: Similar as in optics, the CPA in heat diffusion is achieved at one “frequency” point for a system. The narrow bandwidth is due to the k -dependence of the scattering matrix. As we show in Supplementary Note 1 of the revised SI, the temperature field has a θ -dependence $e^{im\theta}$, where m is an integer. By comparing with the time-dependence of a wave field $e^{i\omega t}$, it turns out that the thermal counterpart of frequency ω is m . In the main text, we focus on the case $m = 1$, because it corresponds to the most common kind of thermal fields. If the “frequency” is changed, the CPA condition should be modified to

$$\kappa = \kappa_0 \frac{r_1^m \mp r_2^m}{r_1^m \pm r_2^m} \quad (3)$$

Thank you for the question on laser-like characteristics. We are also very interested in this exciting possibility. However, note that the time-reversed heat diffusion requires a negative thermal diffusivity (see below), which has not been realized. It thus remains a great challenge to achieve the thermal analogue of lasing effects. We have included the discussion in the revised manuscript (page 8).

(5) An intriguing aspect then is how exactly time reversal symmetry (TRS) is being violated- as indicated on page 6. Firstly, some kind of a “pseudotime” is being used. Is the TRS related to this pseudotime or regular/ordinary time? It has also been rather glibly stated that “Together, the system preserves a parity symmetry which is not obvious from its geometry”? How could such a statement be more clearly justified?

Our reply: We apologize for the confusion. The time-reversal symmetry is related to ordinary time. In heat diffusion, the temperature variance will decay with time, while its time-reversed version should have a growing temperature variance. Therefore, the TRS is always broken in heat transfer. This is also clear from the governing equation

$$\frac{\partial T}{\partial t} = D\nabla^2 T \quad (4)$$

where $D = \kappa/(\rho c_p)$ is thermal diffusivity, ρ is density, and c_p is the specific heat capacity. The time-reversed version of Eq. (4) is

$$\frac{\partial T}{\partial t} = -D\nabla^2 T \quad (5)$$

We see that the TRS is violated, and the thermal diffusivity D is mapped to $-D$. We thank the referee for raising the question. The ordinary time and pseudo time are different, so the original discussion is not very relevant, which has been removed from the revised manuscript.

The parity symmetry indicates the symmetry of the temperature field on $x = \ln r$. We already have

$$\begin{aligned} F_1(x) &= A_1 e^{x-x_1} + B_1 e^{-(x-x_1)} = A_1 e^{x-x_1} + (r_{11}A_1 + t_{12}A_2) e^{-(x-x_1)} \\ F_2(x) &= B_2 e^{x-x_2} + A_2 e^{-(x-x_2)} = (t_{21}A_1 + r_{22}A_2) e^{x-x_2} + A_2 e^{-(x-x_2)} \end{aligned} \quad (6)$$

For parity symmetric scattering matrix, $r_{11} = r_{22}$ and $t_{12} = t_{21}$. The corresponding parity operation is swapping the exterior and interior parts of the object. This is achieved through the following mapping

$$x \leftrightarrow x_1 + x_2 - x, A_1 \leftrightarrow A_2 \quad (7)$$

Under the operation, it turns out

$$F_1(x) \leftrightarrow F_2(x) \quad (8)$$

This symmetry is unobvious because it is based on x which is a logarithmic coordinate, so one cannot directly see it from the temperature field. We have included the discussion in the revised manuscript (page 6).

(6) Also, the authors indicate occasionally that there is “no thermal correspondence” – on page 8, as phase considerations are irrelevant?

Our reply: We are sorry that the discussion on page 8 may be inappropriate. There is a thermal correspondence to the phase in optics. In optics, the phase is reflected in the time-dependence of the EM field, or equivalently in the angle of its complex amplitude. Correspondingly, in our system, the phase is reflected in the θ -dependence of the temperature field. We have added a note to the revised SI (Supplementary Note 3) to discuss the effects of a phase difference α between the input fields. To introduce it, we just need to rotate the direction of the dipole-like input-2 field by an angle $-\alpha$. The input-2 field then has a complex amplitude $A_2 e^{i\alpha}$. From Fig. S1 of the revised SI, we can see that the introduction of a phase difference breaks the CPA condition. We have also performed a new experiment to demonstrate it (Fig. S1d). Moreover, when $\alpha = \pm\pi$, the input is just the eigenvector corresponding to the other nonzero eigenvalue of the scattering matrix. The outgoing fields are maximized at this phase difference, corresponding to the optical

case of constructive interference. We have included the discussion in the revised manuscript (page 8).

(7) In the numerical simulation, what is the reason for the choice of the “representative values”? How do the temperature profiles in Figs. 2e and 2f or Figs. 3c or 3d. accord with a coherent aspect?

Our reply: Thank you for the question. Our method is generally applicable to various values. In numerical simulations, we choose thermal conductivities of common materials such as aluminum and steel, and geometric sizes commonly used in macroscopic heat transfer. The temperature difference is around 5 K, such that the field can be easily generated with heat sinks and measured with an infrared camera.

Under the CPA condition, there should be no outgoing field, so the temperature fields should have the same r -dependence as the input files. In Figs. 2e and 2f, the temperature profiles (scatters) outside the object ($w < -4$ cm and $w > 4$ cm) meet well with the profile of input-1 (red solid lines). Also, those inside the object (-2.5 cm $< w < -1$ cm and 1 cm $< w < 2.5$ cm) meet well with the profile of input-2 (blue solid lines). It is thus confirmed that no outgoing field exists on both sides of the object ($B_1 = B_2 = 0$). Since $B_1 = r_{11}A_1 + t_{12}A_2$ and $B_2 = t_{21}A_1 + r_{22}A_2$, the vanishing of outgoing field outside (inside) the object could be understood as the analogue of destructive interference between the reflected (transmitted) field of input-1 and transmitted (reflected) field of input-2. This coherent aspect is further confirmed by the results in Fig. S1 of the revised SI, where a phase difference breaks the condition and results in outgoing fields. In Fig. 3c, only the temperature profile outside the object meets the profile of the input-1 field. It implies that $Z_1 = 0$. Note that Z_1 not only contains the reflected field of input-1 and the transmitted field of input-2, but also other multiplicatively scattered fields (see Eq. (11) of the main text). Therefore, the one-side CPA effect in Fig. 3c can be understood as the “destructive interference” between all those fields. Similar discussion applies to the results in Fig. 3d. For clarity, we have changed the discussions in the revised manuscript (page 10).

There are also rather sweeping statements such as “Only the complete annihilation of two opposite fields is nontrivial” which are quite confusing in scope. What is also a “one side CPA”? Is there an optical analog?

Our reply: We apologize for the inaccurate statements. In the revised manuscript and SI, we have avoided such qualitative statements, and have discussed the general case with a phase difference between the input fields. We have also defined a TEP flux to quantify the total amount of the outgoing fields. We have included the discussion in the revised manuscript (page 8).

The one-side CPA could be understood as the analogue of unidirectional absorption in optics. Namely, there is no outgoing field at just one side of the object. This is a loosened condition compared to the CPA condition, so one can achieve it by adjusting the input amplitudes for arbitrary thermal conductivity of the object. We have included the discussion in the revised manuscript (page 9).

From the experimental point of view, the results in, e.g., Figures 4c and 4d, are intriguing, in terms of a “dipole-like” characteristic. It is unclear how these results are in accord with the simulations indicated previously. The authors should clearly explain the similarities and differences and why CPA is obvious to them.

Our reply: Thank you for the comments on our experiments. As mentioned above, the CPA in heat diffusion is demonstrated as no outgoing fields on both sides of the object, such that the temperature profiles should have the same r -dependence as the input fields. In particular, the field outside the object should be a linear distribution in the lateral direction, such that the isothermal lines are all vertically oriented. This can be easily checked from the infrared images in Figs. 4c and 4d, which are in accordance with the simulated results in Figs. 2c and 2d. More quantitatively, in Figs. 4e and 4f, the measured results (scatters) outside the object ($w < -4$ cm and $w > 4$ cm) are in consistent with input-1 (red solid lines). Also, those inside the object (-2.5 cm $< w < -1$ cm and 1 cm $< w < 2.5$ cm) are in consistent with input-2 (blue solid lines). The CPA effect is thus verified in the same way as for the simulation results in Figs. 2e and 2f. For clarity, we have changed the discussions in the revised manuscript (page 12).

In summary, considering the lack of a clear definition of a phase, it is difficult to understand the related coherence aspect in the invoked CPA. The use of a pseudotime needs much more clear explanation. In summary, the authors should reconsider the physical premise/s behind their work and truly aim to indicate the utility of their approach for obtaining insights into thermal transport.

Our reply: Thank you for the suggestions. The definition of a phase has been elaborated in our revised SI. As mentioned above, it is reflected in the θ -dependence of the temperature field, and can be written as the angle of the complex amplitude. A phase difference α can be easily realized by rotating the orientation of input-2 (the copper bridge in the experiment) by $-\alpha$. By tuning the phase difference, the system is changed from the CPA state to one with large outgoing fields, demonstrating the coherent aspect. We have also added a Supplementary Note 1 in the revised SI to detailly establish the correspondence between 1D wave scattering and 2D heat diffusion, which justifies the use of pseudo time. The referee’s comments encouraged us to revise and clarify some physical interpretations in our work. We expect that the revised version will clearly demonstrate the novel insights on heat transfer from a scattering perspective.

REVIEWER COMMENTS

Reviewer #1 (Remarks to the Author):

Authors have implemented all comments and i find the paper in publishable form.

Reviewer #2 (Remarks to the Author):

The authors have done a good job of addressing the comments and criticisms raised during the first round of reviewing. In particular, they have introduced new material to clarify the meaning of their "pseudotime", as well as the issue of what is actually "absorbed" in this thermal analogue of "coherent perfect absorption".

In my view, it is still an open question whether this concept will really lead to profitable new avenues of research -- some of the contortions, like the use of an angle variable as a pseudotime, seem a bit limiting. But it is still an interesting idea worth disseminating. I support publication.

One last note: there are various grammar mistakes throughout the manuscript, so I suggest doing a round of thorough checking before final submission.

Reviewer #3 (Remarks to the Author):

In the revised version, the recasting in terms of non-Hermitian physics is interesting. The mathematical aspects are also quite clear. However, the analogy of the thermal conductivity to an "electrical admittance" needs considerable interpretation, especially since the product of ϵ and μ is -1 , from Eqn. (S3). Then, only ϵ and μ , are relevant? Additionally, the invoking of a thermoelectric potential (TEP) seems to imply that the temperature is to be interpreted as BOTH an electric field as well as a magnetic field – which raises questions related to whether appropriate physical correspondence is being made! I feel that the introduction of the thermoelectric potential (TEP) has confused the issue, since one is looking for an electrical counterpart to the CPA based phenomena.

While it is indeed true that the "establishment of the thermal scattering theory is ...challenging" - however, the broad aspect is whether heat transport should be considered in terms of particle scattering and not wave scattering? While the aspect of $x=\ln(r)$ as well as the "scattering" in the " r " and the " $1/r$ " directions is relevant from the Fourier law, the introduction of an orthogonal θ , still seems somewhat superfluous unless a time-dependent heat equation – such as the Cattaneo form is considered?

The persistent issue with the work is the physical relevance of the imaginary time concept and the consideration of heat transport in terms of waves in an alternate "dimension". For instance, it has been indicated that there would be a "periodicity to the pseudotime" (In the response by the authors which is again unclear. For instance, how could time have a periodicity?

The question related to the “imitated momentum” and the wave vector has not been clarified. It would truly help if the authors clarified the physical meaning and the implication behind the statement: $k = -i$. Similarly, the rationale of making an equivalence of the temperature to the electric field in Supplementary Note 1. I am talking about more than just a cursory mathematical equivalence!

The introduction of a phase is somewhat ad hoc and has not been justified. If the temperature field “does not really propagate” how could a phase be meaningful? The units of the flux (f) in Eqn. S15 is confusing. Does this somehow reduce to W/m^2 . If so, how?

In summary, the correspondence between the thermal aspect with the optical terminology is still unclear. The physical rationale and the motivation for a scattering-based approach needs to be clearly explained.

** See Nature Research’s author and referees' website at www.nature.com/authors for information about policies, services and author benefits.

Reply to Reviewer #1:

Authors have implemented all comments and I find the paper in publishable form.

Our reply: Thank you very much for the recommendation.

Reply to Reviewer #2:

The authors have done a good job of addressing the comments and criticisms raised during the first round of reviewing. In particular, they have introduced new material to clarify the meaning of their "pseudotime", as well as the issue of what is actually "absorbed" in this thermal analogue of "coherent perfect absorption".

Our reply: Thank you very much for the positive comments. In this revision, we have further generalized our discussion by using the concept of exergy, a thermodynamic quantity defined as the maximal work that can be extracted from the system. The concept of thermoelectric potential (TEP) coined by us has been avoided. It is now clear that the thermal CPA corresponds to the perfect absorption of the exergy fluxes.

In my view, it is still an open question whether this concept will really lead to profitable new avenues of research -- some of the contortions, like the use of an angle variable as a pseudotime, seem a bit limiting. But it is still an interesting idea worth disseminating. I support publication.

Our reply: Thank you very much for supporting publication. We are glad to see that our idea is found interesting and worth disseminating. We would also like to mention that the use of angle variable as a pseudo time is already applicable to a wide range of structures (as shown in Supplementary Note 4), thanks to our general method for arbitrary shapes. Moreover, the angle variable is certainly not the only choice for the pseudo time. Other alternative candidates also exist, such as periodically varying system parameters (like those used to realize synthetic dimensions). Therefore, we are confident that our concept will lead to profound new researches.

One last note: there are various grammar mistakes throughout the manuscript, so I suggest doing a round of thorough checking before final submission.

Our reply: Thanks for the comments. We have made a thorough check on the grammar of the manuscript.

Reply to Reviewer #3:

In the revised version, the recasting in terms of non-Hermitian physics is interesting. The mathematical aspects are also quite clear. However, the analogy of the thermal conductivity to an “electrical admittance” needs considerable interpretation, especially since the product of ϵ and μ is -1 , from Eqn. (S3). Then, only ϵ and μ , are relevant?

Our reply: Thanks for the positive comments. We agree that the analogy could be further elaborated. Our analogy is based on the forms of the fields in the 1D EM scattering and 2D heat transfer problems. In particular, the matching conditions for the temperature fields are:

$$\begin{aligned} F_1(x_1) &= F(x_1), & F_2(x_2) &= F(x_2) \\ \kappa_0 F_1'(x_1) &= \kappa F'(x_1), & \kappa_0 F_2'(x_2) &= \kappa F'(x_2) \end{aligned} \quad (1)$$

Those for the electric fields are:

$$\begin{aligned} E_1(x_1) &= E(x_1), & E_2(x_2) &= E(x_2) \\ \frac{1}{\mu_0} E_1'(x_1) &= \frac{1}{\mu} E'(x_1), & \frac{1}{\mu_0} E_2'(x_2) &= \frac{1}{\mu} E'(x_2) \end{aligned} \quad (2)$$

The analogy between the first lines of Eqs. (1) and (2) is automatically established through the mapping from $F(x)$ to $E(x)$. For the analogy between the second lines, the explicit forms of the fields (see Eqs. (S5) and (S6) of the SI) should be considered to give

$$F_{1,2}'(x) = ik_0 F_{1,2}(x), \quad F'(x) = ikF(x) \quad (3)$$

where $k_0 = k = -mi$. And

$$E_{1,2}'(x_1) = ik_0 E_{1,2}(x), \quad E'(x) = ikE(x) \quad (4)$$

where $k_0 = (\epsilon_0 \mu_0)^{1/2} \omega$ and $k = (\epsilon \mu)^{1/2} \omega$. The second lines can thus be rewritten as

$$\kappa_0 F_1(x_1) = \kappa F(x_1), \quad \kappa_0 F_2(x_2) = \kappa F(x_2) \quad (5)$$

and

$$\sqrt{\frac{\epsilon_0}{\mu_0}} E_1(x_1) = \sqrt{\frac{\epsilon}{\mu}} E(x_1), \quad \sqrt{\frac{\epsilon_0}{\mu_0}} E_2(x_2) = \sqrt{\frac{\epsilon}{\mu}} E(x_2) \quad (6)$$

The analogy between Eqs. (5) and (6) gives the mapping from the thermal conductivity to the admittance

$$\kappa_0 \rightarrow Y_0 = \sqrt{\frac{\epsilon_0}{\mu_0}}, \quad \kappa \rightarrow Y = \sqrt{\frac{\epsilon}{\mu}} \quad (7)$$

The above derivations can also be summarized by the transfer matrices \mathbf{M} for the temperature fields and \mathbf{M}_E for the electric fields, as in the SI. We also have the matching condition from the governing equations

$$-1 \rightarrow \varepsilon_0 \mu_0, -1 \rightarrow \varepsilon \mu \quad (8)$$

The two conditions on ε and μ in Eqs. (7) and (8) make both ε and μ relevant. We have included the discussion in the revised SI (Supplementary Note 1).

Additionally, the invoking of a thermoelectric potential (TEP) seems to imply that the temperature is to be interpreted as BOTH an electric field as well as a magnetic field – which raises questions related to whether appropriate physical correspondence is being made! I feel that the introduction of the thermoelectric potential (TEP) has confused the issue, since one is looking for an electrical counterpart to the CPA based phenomena.

Our reply: Thank you for the concern on TEP. Basically, we are interested in the temperature difference $T - T_0$ and the thermal energy it carries. In this revision, we have avoided the use of TEP. Instead, we use an existing concept—exergy for our discussion. The exergy is the maximal work that can be extracted from the system using the temperature difference. It turns out that the r - and r^{-1} -components of the temperature field carry exergy fluxes in opposite directions. We have revised the manuscript and SI (Supplementary Note 2) accordingly.

While it is indeed true that the “establishment of the thermal scattering theory is ...challenging” - however, the broad aspect is whether heat transport should be considered in terms of particle scattering and not wave scattering? While the aspect of $x = \ln(r)$ as well as the “scattering” in the “ r ” and the “ $1/r$ ” directions is relevant from the Fourier law, the introduction of an orthogonal θ , still seems somewhat superfluous unless a time-dependent heat equation – such as the Cattaneo form is considered?

Our reply: Thank you for the question. Throughout this work, we only consider heat transfer following Fourier’s law, because heat wave (Cattaneo or other forms) is hardly observable in macroscopic systems at room temperature. Therefore, the heat transfer under consideration is a steady-state pure diffusion. Although at the atomic level, there are scatterings of phonons and electrons, at the macroscopic level temperature field is not regarded as a wave or a particle. Despite the analogy to 1D wave scattering, we always keep in mind that it is essentially a 2D heat transfer process. The thermal scattering theory is proposed as a novel tool to relate different components of a temperature field.

We fully agree with the referee on the relevance of identifying the “ r ” and “ $1/r$ ” components. This is made possible exactly thanks to the introduction of an orthogonal θ -coordinate. Without this auxiliary dimension, the steady-state heat transfer in 1D has solution $T(x) = Ax + B$, which does not separate into two components. Therefore, our method is not superfluous but necessary to construct a scattering theory. It is actually in the time-dependent case that one can also study 1D scattering by considering oscillating heat sources (see our recent work *Phys. Rev. B* **103**, 014307).

The persistent issue with the work is the physical relevance of the imaginary time concept and the consideration of heat transport in terms of waves in an alternate “dimension”. For instance, it has been indicated that there would be a “periodicity to the pseudotime” (In the response by the authors which is again unclear. For instance, how could time have a periodicity?)

Our reply: We are sorry that the referee is still unsatisfied with our considerations. We first answer the question on periodicity. What we meant is simply that the temperature field has a periodicity on the pseudo time θ : $T(x, \theta) = T(x, \theta + 2m\pi)$, thanks to the geometry of the system. It does not mean that the coordinate itself has a periodicity. It is the same as for wave field which is periodic on time t : $E(x, t) = E(x, t + \tau)$, where τ is the period. In that sense, the θ -coordinate plays the same role as time for heat transfer.

In general, making analogies between distinct physical phenomena in different dimensions is a common practice that often leads to novel physical insights or methods. For example, one may regard optical beams as quantum wave functions, because the paraxial equation for the former has the same form as the Schrödinger equation when the spatial coordinate in the propagation direction is regarded as time (*Phys. Rev. Lett.* **100**, 103904). Conversely, one may construct an additional synthetic spatial dimension from the temporal dependence of a lower-dimensional system (*Optica* **5**, 1396-1405). Our work follows the same wisdom. Its physical relevance is evident by providing new tools to control heat transfer (additional heat sources) and revealing the physical meaning of different components of the temperature field (carrying exergy fluxes).

The question related to the “imitated momentum” and the wave vector has not been clarified. It would truly help if the authors clarified the physical meaning and the implication behind the statement: $k = -i$.

Our reply: Thanks for the suggestion. The wavenumber k determines how the temperature varies along the radial direction. The solutions $e^{\pm ikx}$ with $k = -i$ means that the temperature difference monotonically increases or decreases along the radial direction.

Similarly, the rationale of making an equivalence of the temperature to the electric field in Supplementary Note 1. I am talking about more than just a cursory mathematical equivalence!

Our reply: Thank you for the comments. We did not mean that the temperature is equivalent to the electric field. They are two completely different physical quantities. In our approach, the temperature is mapped to the electric field under our analogy, such that there is a *correspondence* between them. This mathematical correspondence is not cursory but rigorous, such that the temperature and electric fields have exactly the same form. The rationale of making this correspondence is to construct a scattering theory for the 2D heat transfer, which provides a systematic way to control heat transfer with additional heat sources.

The introduction of a phase is somewhat ad hoc and has not been justified. If the temperature field “does not really propagate” how could a phase be meaningful?

Our reply: Thanks for the question. The temperature field does not propagate because it does not oscillate in the r -direction. As a result, the concept of phase is indeed missing in its r -dependence. Instead, the phase α of a temperature field is defined through its dependence on the θ -coordinate:

$T(x, \theta) = Ae^x \cos(\theta + \alpha)$. It follows the standard definition of the phase of a periodic function. We have clarified the point in the revised manuscript and SI.

The units of the flux (f) in Eqn. S15 is confusing. Does this somehow reduce to W/m^2 . If so, how?

Our reply: Thank you for the question. The original definition of f is based on the TEP, which is not general enough. In this revision, we study the local exergy density, which has the same unit as energy density. The exergy flux χ_f is defined as

$$\chi_f = \left(1 - \frac{T_0}{T}\right) \mathbf{q} \quad (9)$$

It has the unit W/m^2 , which is the same as that of heat flux.

In summary, the correspondence between the thermal aspect with the optical terminology is still unclear. The physical rationale and the motivation for a scattering-based approach needs to be clearly explained.

Our reply: Thank you for the suggestion. The ultimate goal of making this correspondence and thereby establishing a thermal scattering theory is to find new ways to control heat transfer, which is a fundamental and important problem. What we found is that we could introduce additional heat sources to achieve thermal CPA. This kind of effects is useful for thermal energy utilization and temperature control. The method we proposed is directly inspired by the interference of optical waves. Without introducing the optical terminology and the scattering theory, it is impossible for us to discover and physically understand the mechanism behind the thermal CPA effects.

In addition, the fact that two distinct physical phenomena share the same mathematical structure is itself an interesting and useful discovery. It leads to deeper physical insights. For example, we now understand that the two components in the temperature field carry exergy fluxes in opposite directions. The analogy also provides a useful physical platform to simulate one phenomenon with the other. We have included the discussion in the revised manuscript.

REVIEWER COMMENTS

Reviewer #2 (Remarks to the Author):

In my view, the current manuscript is suitable for acceptance in terms of scientific content. There are still some typos (e.g., "exergy fluxes" (x4), "temperature managements") which should be corrected before publication.

Reviewer #3 (Remarks to the Author):

The authors have now used another completely new methodology, i.e., exergy, to explain the underlying physics. This is confusing again (so sorry, but I am really trying to understand!) but what is the relative contributions of the free energy and entropy here - say, is there any relevance to the pseudo-time approach considered in this work?

Statements such as "The exergy flux enters or leaves a local spot to modify the local exergy density, which is the theoretically maximal work that can be extracted from the system" are confusing and must be revised. Is the CPA related analogy being related to work now?

The related definition of a wavenumber, in the form: $k = -i$, is still confusing. Looking at the cited reference, i.e., ref. 39, the notion of an imaginary wavenumber (as for evanescent waves) is common but typically, it is of the form: $k = -ix$, where x is a real number (as also indicated in ref. 39). What is the real number/related component in this work?

Reply to Reviewer #2:

In my view, the current manuscript is suitable for acceptance in terms of scientific content. There are still some typos (e.g., "exergy fluxes" (x4), "temperature managements") which should be corrected before publication.

Our reply: Thank you very much for supporting the publication. We have carefully checked and corrected the typos. We would like to mention that exergy is not a typo of energy, but a different concept in thermodynamics.

Reply to Reviewer #3:

The authors have now used another completely new methodology, i.e., exergy, to explain the underlying physics. This is confusing again (so sorry, but I am really trying to understand!) but what is the relative contributions of the free energy and entropy here - say, is there any relevance to the pseudo-time approach considered in this work?

Our reply: Thank you for the question. The introduction of exergy is a modification on our previous explanation of the thermal CPA effects based on thermoelectric potential (TEP). It is not a new methodology. Our main methodology is still to make analogies between heat and wave. We are sorry that the referee found the concept of exergy confusing. The reason might be that the referee regards it as a physical interpretation of our *approach*, and thus asks for its relevance to the pseudo-time, which was used in constructing our analogy. However, the exergy is introduced as a physical interpretation of our *effects*. It is analyzed in the standard framework of thermodynamics, without using the concepts in our analogy.

To be more concrete, we would like to review our thoughts in this study. First, we conceived the idea that adding heat sources might be a new way to manipulate heat transfer through an object. To analyze the temperature field surrounding the object, we must establish a thermal scattering theory to calculate the effect of each heat source. However, this is a nontrivial task since the concepts of input and output are missing in the steady-state diffusion. Fortunately, we were able to find the analogy between 1D wave scattering with 2D heat transfer. The approach helps us to decompose the temperature field into different parts. In the approach, one spatial coordinate of the heat transfer system corresponds to the temporal coordinate of the wave system, thus it is called the pseudo time.

Using the analogy-based approach, we designed the models that exhibit thermal CPA effects. Based on the analogy, we see that in the thermal CPA effects there is no outgoing component in the temperature field, corresponding to the perfect absorption in wave scattering. However, this interpretation only answers what is the analogy of the thermal CPA effects in photonics. It does not answer what is the physical meaning of the thermal CPA effects in common thermodynamics. That is why we propose the exergy-based interpretation, which does not rely on making analogy to photonics.

The exergy is a well-defined physical quantity in thermodynamics. In heat transfer, the internal energy density u characterizes the amount of energy at a local point. However, it does not tell how we can make use of the energy. In fact, only a portion of the internal energy is useful and convertible to work when there exists a temperature difference between the local point and the environment, according to the second law of thermodynamics. The exergy is introduced to quantify this “useful energy”. Its definition is given below:

The exergy of a thermodynamic system S in a certain state S_A is the maximum theoretical useful work obtained if S is brought into thermodynamic equilibrium with the environment by means of ideal processes in which the system interacts only with this environment. (Ref. 40)

The above discussion shows that the exergy density χ and the exergy flux χ_f are actually more physically relevant than the internal energy density u and the heat flux q , because they tell how the useful thermal energy is distributed and transferred in the system. Interestingly, our calculation of the exergy flux χ_f shows that this useful thermal energy is perfectly absorbed by the object under the condition of thermal CPA, which gives a concrete physical meaning to our effects.

It should be clear now that the analogy-based approach, or pseudo-time approach, is used to design and solve our models, while the exergy analysis is used to explain our effects. They belong to different steps of this study. The exergy analysis is not supposed to interpret the pseudo time. It was performed in the standard way where the coordinates are the physical space and time, not considered as the pseudo time.

Statements such as "The exergy flux enters or leaves a local spot to modify the local exergy density, which is the theoretically maximal work that can be extracted from the system" are confusing and must be revised. Is the CPA related analogy being related to work now?

Our reply: Thank you for the suggestion. We have completely revised the statement as:

“The exergy is a thermodynamic quantity defined as the maximum useful work a system can do by bringing it into thermodynamic equilibrium with the environment⁴⁰. In our case, the useful work comes from the temperature difference between any local point in the system and the environment⁴¹, meaning that one can extract work by putting a heat engine between them. Therefore, our decomposition of the temperature field gives important information about how the potentially useful thermal energy is distributed and transferred in the system.”

In other words, the exergy is the useful thermal energy stored in the system. According to our exergy analysis, the thermal CPA effect corresponds to the perfect absorption of exergy fluxes. It means that this useful thermal energy is transferred from the heat sources into the object at the maximum efficiency.

The related definition of a wavenumber, in the form: $k = -i$, is still confusing. Looking at the cited reference, i.e., ref. 39, the notion of an imaginary wavenumber (as for evanescent waves) is common but typically, it is of the form: $k = -ix$, where x is a real number (as also indicated in ref. 39). What is the real number/related component in this work?

Our reply: Thank you for the question. The expression for k is actually not defined by us, but directly solved from the heat transfer equations. In our model, the geometry requires that temperature field has the form $T(x, \theta) = \text{Re}[F(x)e^{im\theta}]$, where m is an integer and $F(x)$ is given by equation (S6) of the Supplementary Information (SI)

$$F(x) = Ae^{ikx} + Be^{-ikx} \quad (1)$$

By substituting the solution into the governing equation (S2) in the SI

$$\frac{\partial^2 T}{\partial x^2} = -\frac{\partial^2 T}{\partial \theta^2} \quad (2)$$

we obtain the $k = -mi$. The integer m is the real number in common notions, it characterizes how fast the temperature field varies along the θ -coordinate. In the main text, we focus on the most common case that $m = 1$. That is why the real number appears missing in the expression.

REVIEWERS' COMMENTS

Reviewer #3 (Remarks to the Author):

While the analogy of the 1D wave scattering problem with the 2D heat transfer assumes a pseudotime, the exergy analysis offered by the authors - to explain the effects of the wave scattering uses traditional thermodynamics.

My issue related to the use of exergy in the paper is that it has not been proven that the exergy fluxes are perfectly absorbed. This has just been stated. Moreover, since exergy involves entropy considerations are the authors indicating that thermal CPA corresponds to no net entropy production? This is different than the CPA in photonics which has to involve dissipation.

However, at this point the wave scattering approach for fitting thermal phenomena seems to have been done. Perhaps, this paper could give others ideas on how to proceed further with respect to non-hermitian physics involving thermal systems.

Reply to Reviewer #3:

While the analogy of the 1D wave scattering problem with the 2D heat transfer assumes a pseudotime, the exergy analysis offered by the authors - to explain the effects of the wave scattering uses traditional thermodynamics.

My issue related to the use of exergy in the paper is that it has not been proven that the exergy fluxes are perfectly absorbed. This has just been stated. Moreover, since exergy involves entropy considerations are the authors indicating that thermal CPA corresponds to no net entropy production? This is different than the CPA in photonics which has to involve dissipation.

However, at this point the wave scattering approach force fitting thermal phenomena seems to have been done. Perhaps, this paper could give others ideas on how to proceed further with respect to non-hermitian physics involving thermal systems.

Our reply: Thank you very much for the comments. We would like to clarify that it has been rigorously proven that the exergy fluxes are perfectly absorbed. Based on Supplementary Equations (20) and (21) in Supplementary Note 2, we know that the total exergy flux at the outer boundary ($x = x_1$) of the object is

$$Q_1 = -\frac{2\pi\kappa}{T_0}(A_1^2 - B_1^2) \quad (1)$$

The plus and minus signs of the flux indicate that it is in the direction of r or $-r$. Therefore, the total exergy flux that leaves the object from its outer boundary is the positive part of Q_1

$$\frac{2\pi\kappa}{T_0} B_1^2 \quad (2)$$

Similarly, the total exergy flux at the inner boundary ($x = x_2$) of the object is

$$Q_2 = -\frac{2\pi\kappa}{T_0}(B_2^2 - A_2^2) \quad (3)$$

The total exergy flux that leaves the object from its inner boundary is the negative part of Q_2

$$-\frac{2\pi\kappa}{T_0} B_2^2 \quad (4)$$

In our main text, we have found the rigorous solutions $\kappa = \kappa_{\pm}^*$ for the conditions:

$$B_1 = B_2 = 0 \quad (5)$$

According to Eqs. (2) and (4), it is clear that the solutions exactly indicate zero outgoing exergy flux from the object, or perfect absorption of the exergy flux. Therefore, our claim is not simply stated, but well justified. We have further clarified the statement in Supplementary Note 2.

We agree with the reviewer that the exergy (χ) is related with the entropy (s) through Supplementary Equation (14) in Supplementary Note 2. In addition, the exergy flux (χ_f) is related with the entropy flux (s_f) as:

$$\chi_f = \mathbf{q} + T_0 \mathbf{s}_f \quad (6)$$

However, our conclusion is that the thermal CPA corresponds to no outgoing exergy flux. This condition only implies a relation between the entropy flux and the heat flux through Eq. (6). On the other hand, the exergy generation (χ_g) is related with the entropy generation (s_g) as

$$\chi_g = -T_0 s_g \quad (7)$$

where the entropy generation is

$$s_g = \mathbf{q} \cdot \nabla \left(\frac{1}{T} \right) \quad (8)$$

As long as there are heat flux \mathbf{q} and temperature gradient, both the exergy generation and entropy generation must be nonzero. Therefore, dissipation is always present, which is indeed a necessary ingredient in CPA effects.

We thank the reviewer for indicating the possibility of further works on non-Hermitian physics involving thermal systems, which is indeed a very interesting direction.